# Melatonin: A Potential Regulator of DNA Methylation

**DOI:** 10.3390/antiox12061155

**Published:** 2023-05-25

**Authors:** Kinga Linowiecka, Andrzej T. Slominski, Russel J. Reiter, Markus Böhm, Kerstin Steinbrink, Ralf Paus, Konrad Kleszczyński

**Affiliations:** 1Department of Human Biology, Faculty of Biological and Veterinary Sciences, Nicolaus Copernicus University, Lwowska 1, 87-100 Toruń, Poland; 2Dr. Phillip Frost Department of Dermatology & Cutaneous Surgery, University of Miami Miller School of Medicine, Miami, FL 33125, USA; 3Department of Dermatology, Comprehensive Cancer Center, University of Alabama at Birmingham, Birmingham, AL 35294, USA; 4Pathology and Laboratory Medicine Service, VA Medical Center, Birmingham, AL 35294, USA; 5Department of Cell Systems and Anatomy, UT Health, Long School of Medicine, San Antonio, TX 78229, USA; 6Department of Dermatology, University of Münster, Von-Esmarch-Str. 58, 48149 Münster, Germany

**Keywords:** melatonin, DNA methylation, active DNA demethylation, DNA methyltransferases, ten-eleven translocation proteins, epigenetics

## Abstract

The pineal gland-derived indoleamine hormone, melatonin, regulates multiple cellular processes, ranging from chronobiology, proliferation, apoptosis, and oxidative damage to pigmentation, immune regulation, and mitochondrial metabolism. While melatonin is best known as a master regulator of the circadian rhythm, previous studies also have revealed connections between circadian cycle disruption and genomic instability, including epigenetic changes in the pattern of DNA methylation. For example, melatonin secretion is associated with differential circadian gene methylation in night shift workers and the regulation of genomic methylation during embryonic development, and there is accumulating evidence that melatonin can modify DNA methylation. Since the latter one impacts cancer initiation, and also, non-malignant diseases development, and that targeting DNA methylation has become a novel intervention target in clinical therapy, this review discusses the potential role of melatonin as an under-investigated candidate epigenetic regulator, namely by modulating DNA methylation via changes in mRNA and the protein expression of DNA methyltransferases (DNMTs) and ten-eleven translocation (TET) proteins. Furthermore, since melatonin may impact changes in the DNA methylation pattern, the authors of the review suggest its possible use in combination therapy with epigenetic drugs as a new anticancer strategy.

## 1. Melatonin Biosynthesis and Function

It has been widely discussed that melatonin has pleiotropic biological effects. This hormone is synthesized in a multistep process from tryptophan via tryptophan hydroxylase (TPH) to 5-hydroxytryptofan (tryptamine). Next, tryptamine is transformed using amino acid decarboxylase (AAD) to 5-hydroxytryptamine (serotonin), with its subsequent conversion to *N*-acetylserotonin (NAS) via aralkylamine *N*-acetyltransferase (AANAT). NAS is the immediate precursor of melatonin (*N*-acetyl-5-hydroxytryptmine), and this step is conducted using hydroxyindole-*O*-methyltransferase (HIOMT) (Figure 1) [1,2].

Due to its chemical structure, melatonin is a lipophilic compound that can easily penetrate biological membranes, thus defending enzymes, proteins, lipids and DNA from oxidative damage [3,4]. It has been shown to play a key role in the protection of cellular organelles, such as nuclei or mitochondria. This indoleamine hormone is synthesized and secreted in a circadian manner by the pineal gland. Its rhythmic secretion affects the sleep/wake cycle [1,5]. In addition, melatonin is produced in the nervous system, as well as in numerous peripheral organs, including the bone marrow, lymphocytes, eyes, gastrointestinal tract [6,7,8,9,10,11,12], and human [13,14,15,16] and rodent skin [17,18]. It exerts anti-oxidative actions by restoring UVB-induced oxidative stress and mitochondrial alterations [4,19], anti-inflammatory effects by decreasing the expression of heat shock protein 70 and pro-inflammatory cytokines [20], or anti-apoptotic properties; thus, melatonin and its metabolites significantly protect against DNA damage via the inhibition of 8-OH-deoxyguanosine formation and the enhancement of p53, as well as in promoting the expression of repair and anti-apoptotic proteins [13,21,22]. These pleiotropic effects are mediated by ubiquitously distributed cell membrane receptors (MT1 and MT2), and they have an impact on members of the RZR/ROR nuclear receptor family, e.g., retinoic acid-related orphan receptor α (RORα), and on membrane receptor-independent mechanisms [11,23,24,25,26].

The anti-oxidative attributes of melatonin are associated with its ability to decrease the number of reactive oxygen species (ROS) or reactive nitrogen species (RNS) involved in the activation of antioxidant enzymes [27]. The latter way requires transcription factor—nuclear erythroid factor 2 (Nrf2)—to play a specific role. Under physiological conditions, Nrf2 is ubiquitinated via ligase E3 complex Keap1-Cul3-Rbx1, which is followed by its proteasomal degradation. However, under oxidative stress, Nrf2 degradation is suppressed, allowing for its translocation to the nucleus and binding to antioxidant response element 2 (ARE) [28,29], which is located in the regulatory regions of genes encoding numerous antioxidant enzymes, including superoxide dismutase (Mn-SOD, Cu/Zn-SOD), catalase (CAT), *γ*-glutamylcysteine synthetase (*γ*-GCS), heme oxygenase-1 (HO-1), NADPH quinone dehydrogenase-1 (NQO1), or glutathione peroxidase (GPx) [4,19,30,31,32,33]. Apart from this, melatonin affects Nrf2 via the inhibition of degradation and the promotion of its nuclear translocation [29,31,34]. Moreover, a recent in vitro study revealed that 1 nM melatonin can effectively stimulate Nrf2 via an increase in the expression levels of antioxidant and proteasome genes linked with the Keap1-Nrf2-ARE pathway [28].

Earlier in vitro and in vivo studies revealed that melatonin can inhibit cancer cell growth. Melatonin regulates the immunosurveillance of cancer cells, via stimulating natural killer (NK) cells, monocytes, leukocytes, interleukins, and interferon-*γ*, as well as activating the cytokine system and cytotoxic activity (Figure 2) [35]. Melatonin reduces the rates of cancer initiation, progression, metastasis and migration [36,37,38,39], angiogenesis, and inflammation [40,41,42]. On the other hand, melatonin can regulate the immune system [43,44] and oxidative stress [19,30,38,45,46,47,48], thereby repressing many malignancies [49].

## 2. Alterations in DNA Methylation

### 2.1. DNA Methylation

DNA methylation results from the attachment of a methyl group to the fifth carbon atom of cytosine in DNA [50]. DNA methylation takes part in fundamental processes in an organism. Under physiological conditions, DNA methylation patterns are not random, and they are controlled and tissue-specific [51]. The process of DNA methylation is driven by DNA methyltransferases (DNMTs), which have the ability to transfer a methyl group from *S*-adenosyl-*L*-methionine to a DNA base—cytosine [52]. There are two primary means of DNA methylation: maintenance methylation driven by DNMT1 [53], and de novo methylation caused by DNMT3A and DNMT3B [54] (Figure 3).

The presence of modified cytosine in promoter regions can affect gene expression. As a general rule, the promoter regions of active genes are hypomethylated, while silenced genes are typically hypermethylated [55]. There are two plausible means of gene expression regulation driven by changes in DNA methylation machinery. Regarding the first of these, 5-mC in DNA is an obstacle for transcription factors by the inhibition of RNA polymerase II binding and the following transcription complexes formation [56]. Recent research revealed, however, that some of transcription factors are dependent of methylated CpG islands; hence, they can interact with hypermethylated DNA [57]. The second one involves 5-mC interaction with specific proteins that recognize the methylated bases of CpG islands—methyl-CpG binding proteins (MeCPs). MeCPs are a heterogeneous group that contain a methyl-CpG binding domain (MBD) [58]. They regulate transcription process through the inhibition of transcription factor binding to DNA or through interaction with other proteins, leading to silencing via heterochromatin formation, which is an inactive form of chromatin [59].

### 2.2. Active DNA Demethylation

A milestone in understanding the dynamic changes in the epigenome was the discovery of ten-eleven translocation (TET) enzymes (TET1, TET2, and TET3). In 2009, Tahiliani et al. [60] demonstrated that TET enzymes (initially regarded as chromosomal translocations in genes *MLL* and *LCX* (t(10;11)(q22;q23)) in leukemia patients [61,62] have the ability to change the DNA methylation pattern [60]. TET enzymes cause the oxidation of 5-mC to 5-hydroxymethylcytosine (5-hmC), leading to DNA demethylation [60]. Subsequent studies revealed that 5-hmC can be further hydroxylated to 5-formylcytosine (5-fC) and 5-carboxylcytosine (5-caC) [63]. Modified cytosines (5-fC or 5-caC) are recognized and excised by thymine DNA glycosylase (TDG), a key enzyme belonging to base excision repair (BER) machinery. As a result, 5-fC or 5-caC are replaced by unmodified cytosine; thus, TDG supports the active DNA demethylation process [64,65] (Figure 4). The loss of methylation marks (5-mC or 5-hmC) can passively occur as a result of mitosis. However, active DNA demethylation is a dynamic process that is independent of the cell cycle.

Apart from the direct regulation of the 5-mC level, TETs can also interact with other proteins, thus affecting DNA methylation. A study on mice embryonic cells indicated that TET1 can attach to Polycomb repressive complex 2 (PRC2) and impact the reduction of 5-hmC in promoter regions, followed by gene repression [66]. It was suggested that it is one of the mechanisms that occurs during cell differentiation, as it was detected in the promoter regions of developmental genes [67]. The association between TET1 and Sin3a—a multifunctional transcription regulator—was also indicated as one of the processes involved in the development and reprogramming associated with transcription inhibition [68,69]. TET proteins can have an interplay with O-GlcNAc transferase (OGT). This interaction can positively influence stem cell production in *tet* mutant zebrafish embryos. Moreover, it can affect the TET2 and 5-mC levels [70].

Modifications in the DNA methylation profile may be a trigger for transcriptional changes, including the activation or silencing of oncogenes and suppressor genes contributing to cell cycle changes, apoptosis inhibition, and the uncontrolled growth of cells, followed by carcinogenesis [71,72,73]. Global DNA hypomethylation is a characteristic hallmark of cancer [74,75], which manifests more frequently in the advanced stages of carcinogenesis [76,77,78,79,80]. In addition, another commonly observed phenomenon in cancer cells is the hypermethylation of CpG islands located in the promoter regions of genes [75]. Many reports have documented that the balance between DNA methylation and demethylation is disturbed in a variety of tumors, as well as TET and DNMT expression [74,81,82,83]. Most changes in TET and DNMT activity are not associated with genetic mutations, indicating that other factors may be responsible for these changes. Based on published data, it is suggested that melatonin may be one of the chemicals that plays a significant role in epigenetics and maintaining the DNA methylation pattern.

## 3. Melatonin and Its Role in DNA Methylation

### 3.1. Melatonin Regulates the Expression of DNMTs

The wide range of the biological effects of melatonin suggests that this indoleamine has a key function in modulating gene expression, including genes related to epigenetic processes. It was demonstrated that melatonin can impact the DNA methylation level, thus stimulating cell differentiation [84]. Additionally, melatonin significantly decreases the methylation level (5-mC) via downregulating the expression of *DNMT1* and *MeCP2*. Of note, the expression levels of *DNMT3A* and *DNMT3B* were not affected by the addition of melatonin itself [84]. On this basis, it may be assumed that melatonin exerts a greater impact on the maintenance of DNA methylation occurring during DNA replication. Herein, a question should be stated: how does melatonin influence DNA methyltransferases? One of the answers was considered by Korkmaz and Reiter [85], and the authors hypothesized that based on their structural resemblance, melatonin and/or its derivatives might potentially serve as DNMT inhibitors via impacting DNMTs transcription or via binding to their catalytic center [85] (Figure 5).

### 3.2. Melatonin and Its Possible Impact on TET Proteins

Recent studies imply that the melatoninergic regulation of DNA methylation, which is followed by changes in gene expression, is not only associated with its impact on DNMTs, but also with other pathways possibly controlled by this indoleamine. A lot of the DNA methylation status in the genome stems from DNA demethylation driven by TET proteins [86]. Since melatonin influences DNMTs expression [84], it seems plausible that this compound may be a potential candidate that influences active DNA demethylation as well. Indeed, mouse embryos with knocked-out aralkylamine *N*-acetyltransferase (AANAT), which is the rate-limiting enzyme in melatonin biosynthesis, resulted in a decrease in TET2 expression and in changes in DNA methylation. Moreover, this process was reversible after melatonin supplementation [87]. Hence, the authors proposed that physiological embryo development requires the secretion of melatonin, as well as TET2 expression, to preserve the DNA methylation pattern and to achieve correct cellular differentiation [87]. Melatonin’s effect on TET2 upregulation was also shown in an in vitro study, where melatonin supplementation led to an increase in TET2, followed by the methylation inhibition of ubiquinol-cytochrome c reductase core protein 1 (UQCRC1), which eventually resulted in improved mitochondrial function [88]. Melatonin was also successfully used to inhibit TET1 expression and the DNA demethylation of metabotropic glutamine receptor subtype 5 (mGluR5), which is implicated in neuropathic pain [89]. The effect of the downregulation of TET1, followed by the decrease in the 5-hmC level at the position of the mGluR5 promoter, was demonstrated via the interaction of melatonin with MT2 receptor [89]. Moreover, melatonin can also significantly increase the *TET1* and *TET3* expression levels after exposure of mycotoxin Enniatin B1 (EB1) to porcine embryos [90]. This was consistent in another study on bovine oocytes, where melatonin supplementation significantly increased the TET1, TET3, and TET2 mRNA and protein levels [91]. It seems that melatonin can differently impact the regulation of TET1, TET2, and TET3 expression. However, it needs to be emphasized that TETs differ significantly in terms of expression during cell development and differentiation [92]. Moreover, DNA demethylation may be not driven the same way by all TET proteins. It was implied that hydroxylation of 5-mC to 5-hmC can be led by all TET proteins; however, the subsequent oxidation of 5-hmC into 5-fC and/or into 5-caC may only be driven by TET2 or TET3 [93].

These data suggest the possibility that melatonin may influence TET gene expression; although, the mechanism remains unknown. One explanation could be the association between melatonin and mitochondrial metabolism, especially in terms of tricarboxylic acid cycle metabolites. As it was demonstrated in mice study, melatonin plays a significant role in increasing the α-ketoglutarate (α-KG, 2-oxoglutarate) level, which is followed by TET-mediated DNA demethylation [94]. TET proteins, as members of 2-oxoglutarate and Fe^2+^-dependent dioxygenases (2-OGDD), require 2-oxoglutarate as a cofactor to maintain their catalytic activity [95,96] (Figure 5). Thus, this action of melatonin may be of importance in the regulation of DNA methylation due to its potential impact on TET expression.

### 3.3. Melatonin and DNA Methylation Changes under Artificial Light at Night

Melatonin synthesis is disrupted under artificial light at night (ALAN) [97]. Pineal melatonin synthesis is regulated by intrinsically photosensitive retinal ganglion cells containing photopigment melanopsin, which are particularly sensitive to exposure to short-wavelength light [98]. Even brief exposition [99] to ALAN may result in the suppression of melatonin secretion. Previous studies demonstrated that reduced melatonin levels provoked via ALAN may be involved in increased cancer incidence, especially hormone-dependent cancers (extensively reviewed in [100]). Further studies revealed that ALAN may be responsible not only for the increase in tumor growth, but also for developing global DNA hypomethylation in the tumors of BALB/c short-day-acclimated mice inoculated with 4T1 breast cancer cells [101]. The same mouse model was also used to detect lower levels of DNMT, along with global DNA methylation, after repeated exposure to ALAN [102]. Interestingly, a previous study documented that the ALAN wavelength is critical for evoking changes in DNA methylation status. A shorter wavelength, up to 500 nm, which is equivalent to that of normal blue LED light, is correlated with tumor development, melatonin suppression, and global DNA methylation. Exposition to longer wavelengths clearly has a weaker impact on melatonin suppression and the DNA methylation status [103]. In each of these experimental studies, melatonin administration significantly diminished ALAN effects on tumor growth, as well as on the DNA methylation level [101,102,103]. Changes in the DNA methylation status after ALAN exposure were also observed in rats lacking malignancies [104]. Moreover, it was previously reported that melatonin can impact cancer metabolism, possibly via epigenetic mechanisms [105,106]. In line with the aforementioned studies, it was demonstrated that ALAN exposure is associated with DNA hypomethylation in pancreatic tissue, as well as with lower levels of glucose and insulin in rats, suggesting its significant effect on metabolic responses [104]. In addition, a recent study indicated that ALAN exposure, particularly among elderly patients, is associated with diabetes, cardiovascular diseases, and obesity incidence [107]. Furthermore, the hypermethylation of *MTNR1B* was recently suggested as a novel epigenetic marker for atherosclerosis [108], which is considered to be a metabolic disease. Taken together, these studies demonstrate the close correlation between melatonin function and the status of DNA methylation. There is a potential role for melatonin in promoting alterations of DNA methylation of metabolic- and carcinogenic-related genes. It is suggested that the association between these factors should be investigated in future studies, as exposition to artificial light at night is acknowledged to be a cancer risk factor [109].

### 3.4. Melatonin and Circadian Clock Epigenetics

The latest research has highlighted that changes in melatonin synthesis and the circadian rhythm are related to cancer development [110,111,112]. There is a relationship between disturbances in the circadian rhythm and genomic instability [113,114]. Moreover, previous research revealed that the maintenance of circadian rhythmicity also involves epigenetic mechanisms [115,116]. Furthermore, the hypermethylation of promotor regions of circadian-cycle-dependent genes was observed in breast cancer cells [114], and melanoma [117], pan-renal cell carcinoma [118], cervical and esophageal cancer [119], and hepatocellular carcinoma cells [120].

The studies involving night shift workers indicated that melatonin secretion is associated with distinctive methylation pattern of genes implicated in the circadian cycle [112,121]. A cross-sectional study, including nurses and midwives who worked day shifts and rotating night shifts, showed differential blood-based DNA methylation statuses in the circadian genes in the latter group [121]. The hypomethylation of promoters was demonstrated by the CLOCK genes, *PER1* and *PER2* [121], which play an important role in regulation during sleep and sleep disorders [122]. Interestingly, hypomethylation status of rotating-shift workers was associated with the reduced methylation of selected circadian genes promoters [121]. A recent study involving female night shift workers in Canada indicated that promoters of circadian genes, *PER3* (a member of *PERIOD* genes, along with *PER1* and *PER2*) and *MTNR1A* (encoding MT1 melatonin receptor), are hypomethylated [112]. Changes in the DNA methylation of circadian genes have also been associated with the differential secretion of melatonin: night shift workers had increased levels of melatonin and peak melatonin secretion was reached later [112]. Furthermore, a large cohort study in Finland revealed the presence of a single-nucleotide polymorphism (SNP), rs12506228, downstream of the *MT1* gene in the vast majority of shift workers. As shown in the study, individuals carrying rs12506228 had significantly lower levels of *MT1,* which is linked to hypermethylation of the regulatory region of *MT1* [123].

### 3.5. Melatonin and Methylation during Cell Development and Differentiation

It has emerged in the literature that melatonin can also regulate methylation patterns during cell development and differentiation. The study involving breeding hamsters indicated that winter-like melatonin significantly decreased the expression of DNA methyltransferases in hypothalamus, followed by the hypomethylation of the *dio3* promoter responsible for gonadal regression [124]. In a mouse study, females treated with decabromodiphenyl ethane (DBDPE)—a flame retardant, proven to have an impact on oocyte toxicity—showed the beneficial effect of melatonin treatment on preimplantation embryo development via the reduction of the number of ROS and the decrease in DNA methylation [68]. Similarly, a favorable effect of melatonin was observed on mice oocytes treated with mycotoxin deoxynivalenol (DON). Exposure of melatonin to mice oocytes during maturation decreased the level of DNA methylation evoked via the previous treatment with DON [125]. Furthermore, porcine embryos exposed to another mycotoxin, EB1, demonstrated alterations in the expression of genes involved in DNA methylation and demethylation, followed by changes in the DNA methylation pattern and the inhibition of embryos development. After melatonin supplementation, methylation changes were mitigated [90]. Melatonin can also promote the development of cloned embryos via an increase in blastocysts formation, the inhibition of embryos apoptosis, and the enhancement of nuclear remodeling [126]. In addition, it was demonstrated that melatonin’s impact on embryos development is associated with its influence on DNA methylation reprogramming. Melatonin can promote changes in the DNA methylation of pluripotency and tissue-specific genes, leading to the improvement of the embryo developmental progress [126]. A recent study reveal that mice embryos have the ability to gradually synthesize melatonin during development. However, the knockdown of rate-limiting enzyme in melatonin synthesis resulted in DNA hypomethylation in blastocysts, followed by the inhibition of further development [87]. Melatonin was also found to have an impact on stem cell differentiation. A recent in vitro study on human dental pulp cells (hDPCs) indicated that melatonin can promote the odontogenic differentiation of hDPCs, with the simultaneous downregulation of global DNA methylation and MeCP [84]. It needs to be emphasized that the loss of methylation stimulates stem cell differentiation, whereas hypermethylation promotes the stemness of cells [127,128]. Moreover, the expression level of *DNMT1* was decreased; however, this change was melatonin-independent [84]. Thus, the mechanism underlying hDPCs differentiation after melatonin supplementation is related to a specific interaction between the proteins involved in global DNA methylation changes.

The aforementioned research suggest that melatonin secretion or supplementation can alter the DNA methylation pattern. Amongst the epigenetics mechanisms, changes in the DNA methylation pattern are the most often described ones, and 5-mC is regarded as a central epigenetic marker [129]. We hypothesize that, apart from many other functions, melatonin may also drive DNA methylation changes. Therefore, the exploration of the potential role of melatonin in DNA methylation changes may contribute to a better understanding of the underlying mechanisms of cancer pathogenesis. Furthermore, given the fact that epigenetic modifications are reversible [130,131,132,133], the modulation of these changes is emerging as a therapeutic intervention. Novel developments in this field have increased the need for inventing new drugs or combined therapies for use in cancer treatment on the epigenome.

## 4. Melatonin as a Part of Combined Epigenetic Therapy

### 4.1. Epigenetic Drugs Limitations

There are several drugs that modulate DNA methylation. The first agents that were proven to exert demethylation activity were the nucleoside derivatives: 5-azacytidine (5-AZA) and its analog—5-aza-2-deoxycytidine (decitabine, DAC) [134]. These compounds are hypomethylating agents, as they have the ability to be incorporated into DNA, instead of cytosine. This results in the arresting of DNMTs due to covalent bond formation, followed by the inhibition of DNMT function [135]. Moreover, these bonds disturb DNA functions, leading to the initiation of DNA damage signaling and DNMTs degradation. The subsequent round of DNA replication is, therefore, associated with the loss of DNA methylation markers [135,136,137]. Despite their clinical success in comparison to that of conventional anti-cancer therapy [138,139,140,141,142], the use of DNMT inhibitors has a number of limitations. Several studies have shown that the treatment responses among patients receiving these hypomethylating agents vary widely. The largest discrepancies are associated with the transcriptional responses to demethylation and the activation of previously silenced promotors after treatment [143,144,145,146,147]. The second limitation of these agents is the instability of 5-AZA and DAC in alkaline and acidic solutions [148,149]. These drugs are only stable in neutral solutions at temperatures relevant for routine clinical use and maintenance, with the preservation of their ability to prevent DNA methylation [135]. Moreover, the activity of azanucleosides is based not only on methylation inhibition, but also on dose-related toxicity, which limits their usefulness [150]. Both, 5-AZA and DAC exhibit significant cytotoxicity at the highest tolerated concentrations in chronic myeloid leukemia cell lines, as well in patients with hematopoietic malignancies [151,152]. Furthermore, prolonged treatment with 5-AZA and DAC (48 h) has a more significant impact on DNMT1 depletion, the induction of DNA damage, and the upregulation of gene expression [153]. Both drugs, however, have relatively short substance elimination half-lives [154,155]. At present, 5-azacytidyne and 5-aza-2-deoxycytidine are the only DNMT inhibitors that have been approved for the treatment of hematological malignancies, including acute myeloid leukemia (AML), chronic myelomonocytic leukemia (CMML), and myelodysplastic disorders (MDS) [156,157]. Currently, there are ongoing clinical trials investigating the plausible use of DNMT inhibitors for the treatment of solid tumors [158]. In fact, the limitations of studies on DNMT inhibitors have raised many questions in need for the further investigation of novel DNA methylation inhibitors or combined therapies relative to their potential minimal toxicity and improved pharmacological properties.

### 4.2. Melatonin’s Potential in Combined Therapy

In line with exploring new combined anticancer therapies, some experimental and clinical research has indicated the beneficial effect of melatonin when it is combined with conventional chemo- or radiotherapy. A recent meta-analysis of sixteen trials, in which a combined treatment with melatonin and chemotherapy was used, revealed that patients who received melatonin with other therapies had better outcomes and survival rates, as well as less frequent side effects, than the patients who did not receive supplemental melatonin did [159]. Interestingly, the most striking result from this meta-analysis is that the same beneficial effects of combined chemotherapy with melatonin were noticed in patients with multiple types of cancer. Furthermore, in vitro studies demonstrated that melatonin can enhance the effect of 5-fluorouracil in colon cancer cells via the activation of caspase-9 and caspase-3 in the mitochondrial apoptotic pathway [160], increasing oxidative stress, downregulating genes and proteins related to apoptosis resistance [161], and inhibiting NF-κB/iNOS and PI3K/AKT signaling pathways [162]. Further evidence for the augmenting effect of melatonin on the anticancer activity of conventional chemotherapy comes from studies of doxorubicin [163,164] and paclitaxel [165].

In light of the recent findings that melatonin combined with conventional anti-cancer therapy could limit the adverse effect of chemotherapy and increase its therapeutic efficacy, including reducing drug resistance [159,160,161,162,166,167], its simultaneous use with DNMT inhibitors or other epigenetic drugs may be a more efficient approach to avoid the limitations of epigenetic drugs. A growing number of clinical trials has investigated the plausible favorable effects of therapies when anticancer drugs are combined with DNMT inhibitors. Currently, 255 clinical trials for 5-AZA with chemotherapy and 175 clinical trials for DAC with chemotherapy have already been registered (clinicaltrials.gov). An in vitro study by Hartung et al. [168] demonstrated that treatment of rat C6 glioma cells with 5-AZA significantly upregulated the expression of the *MT1* melatonin receptor. Epigenetic silencing via the hypermethylation of CpG islands of *MT1* gene in oral squamous cell carcinoma (OSCC) cell lines was mitigated after a treatment with 5-AZA [169]. Notably, it was previously reported that *MT1* loss is associated with a worse prognosis in oral squamous cell cancer [169], triple-negative breast cancer [170,171], and in leiomyosarcoma patients [172]. Moreover, increased *MT1* expression has a favorable effect on the resensitization of breast cancer cells to paclitaxel with the concomitant upregulation of tumor suppressor genes [173] (Table 1). This is in accordance with earlier studies, where the association between MT1 and cancer incidence was reviewed by Hill et al. [174]. The decrease in melatonin and MT1, which is associated with age, was suggested to be one of the risk factors of breast tumor growth [174]. Moreover, not only silencing itself, but also specific methylation pattern of MT1 is significant. A recent study by Lesicka et al. [175] revealed that the profiles of MT1 methylation were significantly different among patients with depression in comparison to those of healthy subjects. In addition, specific single nucleotide polymorphism (SNP) variants in *MTNR1B* combined with *CDKN2A* and *MGMT* genes hypermethylation were found to cause worse 5-year overall survival in colorectal cancer patients, suggesting significant role of SNP variants of *MTNR1B* in cancer pathogenesis [176]. A particular SNP variant of the *MT1* gene was also detected downstream of *MT1* [123]. In silico analyses demonstrated that this variant may be responsible for the silencing of *MT1* in the brain via DNA methylation. Moreover, authors suggested that the mentioned SNP variant is related to the exhaustion of shift workers, potentially leading to a decrease in melatonin signaling and an increase in cancer incidence in this group [123].

### 4.3. Melatonin Plays Significant Role in Changing of DNA Methylation Pattern of Cancer-Related Genes

Lee et al. [177] who studied the characteristics of DNA methylation profiles in breast cancer cells (MCF-7), identified several genes candidates as those affected by melatonin exposure. The authors analyzed genes which mRNA expression level was negatively correlated with the DNA methylation status. After a treatment with melatonin, MCF-7 cells displayed the significant upregulation of tumor suppressor gene (GPC3), as well as the downregulation of oncogenic genes (EGR3 and POU4F2-Brn-3b) [177]. Another study on brain tumor stem cells and glioma cells (A172) proved that melatonin in combination with chemotherapeutic agents influences the hypermethylation of the promoter of the ABCG2/BCRP gene responsible for multidrug resistance and tumor recurrence [178] (Table 1). These findings suggest the novel anti-cancer activity of melatonin. It is, therefore, likely that melatonin may be involved in changing the DNA methylation status of cancer-related genes.

**Table 1 antioxidants-12-01155-t001:** Recent findings promoting melatonin as a methylation modulator.

Studied Cell Line/Mice Model	Main Finding	References
Oral cell carcinoma cell lines	Hypermethylation of CpG islands in *MT1* gene restored after treatment with 5-AZA	[169]
Rat C6 glioma cells	Treatment with 5-AZA increases expression of *MT1* gene	[168]
Porcine oocytes	Treatment with melatonin increased level of 5-mC and DNMT in prolonged culture	[179]
4T1 mouse breast cancer model	Melatonin can mitigate the changes in DNMT activity and global DNA methylation level in mice exposed to artificial light at night	[102,103]
Brain tumor stem cell, A172 malignant glioma tumor cells	Melatonin treatment increases methylation of genes ABCG2/BCRP associated with multidrug resistance	[178]
MCF-7 mice breast cancer model	Melatonin treatment increases the favorable effect of paclitaxel by impact on IL-6, STAT3 and DNMT1 gene expression	[173]
MCF-7 breast cancer cells	Supplementation of melatonin increases expression of tumor suppressor gene and decreases expression of oncogenic genes possibly by impacting on methylation level	[177]

It should be emphasized that melatonin itself may impact DNMTs via regulating mRNA or protein expression [84] (discussed above). Furthermore, alterations in the methylation of promoters of circadian rhythm genes may also impact the sensitivity of anticancer drugs [118]. Hence, it is hypothesized that DNMT inhibitors improve the therapeutic potential of melatonin and that the improvement may be substantial. Future studies on the therapeutic efficacy of these agents in combination are, therefore, needed.

## 5. Conclusions and Future Perspectives

A large and growing body of literature has investigated melatonin as a promising candidate for cancer prevention or treatments. Herein, we discussed melatonin’s potential to modulate DNA methylation. Since night shift work is associated with the hypermethylation of genes involved in the circadian biorhythm [180], which is also involved in changes in melatonin secretion [112], it is plausible that melatonin may also directly change DNA methylation. Given the fact that DNA methylation modifications are the most prevalent epigenetic processes [181], melatonin may affect the regulation of gene expression. Additionally, well-described melatonin roles, such as anti-apoptotic or anti-oxidative molecule ones, can be also related to alterations in the DNA methylation pattern. Previous studies have noted the importance of changes in DNA methylation pattern in genes related to apoptosis in cancer cells undergoing this process [182,183,184]. Moreover, a recent in vivo study revealed that mice exposed to formaldehyde, known for provoking oxidative stress and inflammation, showed a more obvious trend toward global DNA hypermethylation compared to that of the formaldehyde-treated group due to simultaneous melatonin supplementation [185]. Furthermore, the treatment of cumulus cells of prepubertal lambs with melatonin was shown to decrease the methylation level of promoter regions of SOD, GPx, and CAT genes [186]. Therefore, it seems plausible that the anti-oxidative effect of melatonin is also associated with epigenetic changes in the genes involved in oxidative homeostasis.

Lastly, due to the fact that epigenetic alterations are eminently reversible [130,131,132], they have emerged as attractive targets for cancer therapy. However, the currently used DNMT inhibitors have limitations (an inconsistent demethylation response and high toxicity level during prolonged use), which can be potentially overcome by using combined anti-cancer therapies. Accordingly, melatonin has beneficial effects when it is combined with chemotherapy [159,160,161,162,166,167], and it can regulate the expression of DNMT [84] or TET [87,89]. Therefore, it may also be more efficient when it is added to epigenetic therapy. Hence, the administration of melatonin, along with DNMT inhibitors in anticancer therapy, especially in tumors where expression of MT receptors is associated with a better prognosis [169,170,171,172], should be, therefore, under consideration at both experimental and clinical levels.

## Figures and Tables

**Figure 1 antioxidants-12-01155-f001:**
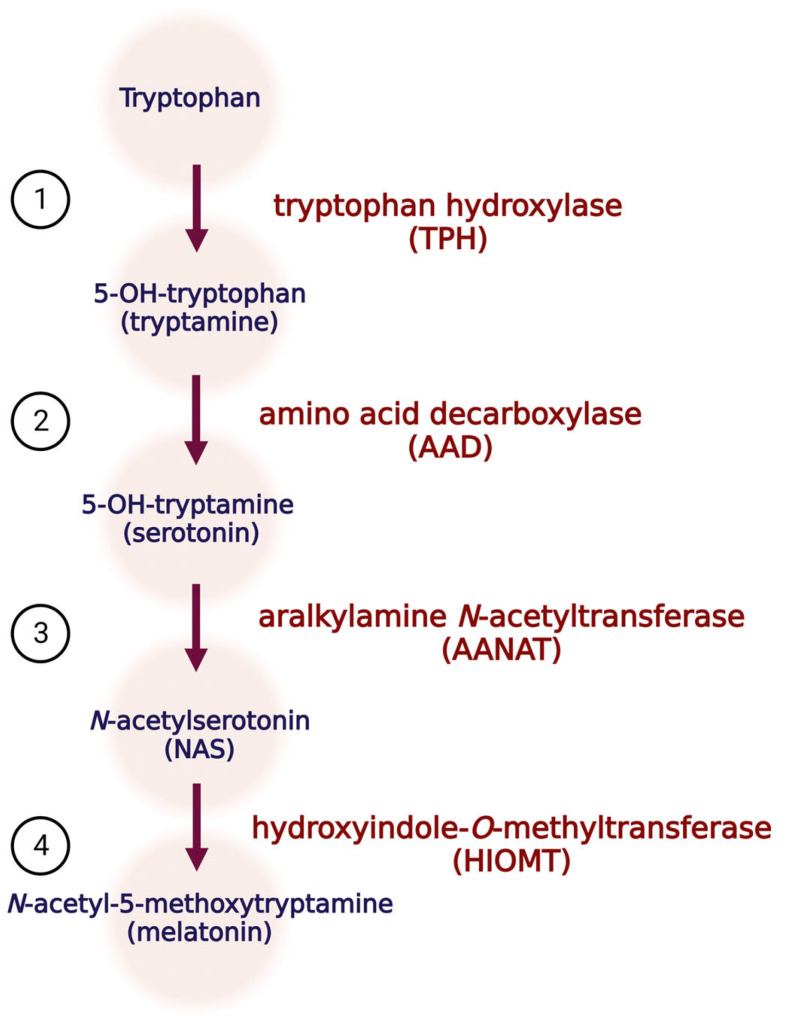
Melatonin synthesis pathway. Abbreviations: TPH: tryptophan hydroxylase; AAD: amino acid decarboxylase; AANAT: aralkylamine *N*-acetyltransferase; HIOMT: hydroxyindole-*O*-methyltransferase; NAS: *N*-acetyloserotonin.

**Figure 2 antioxidants-12-01155-f002:**
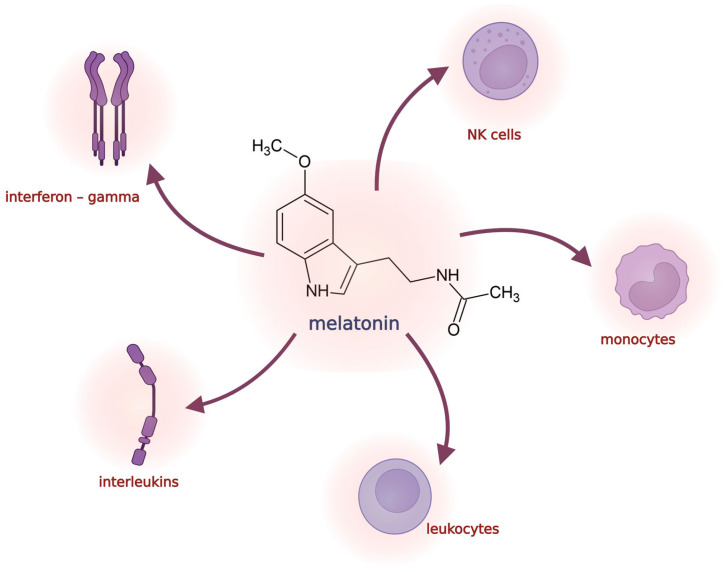
Melatonin stimulates cells of the immune system, NK cells, monocytes, leukocytes, interleukins, and interferon gamma to invoke immune responses triggering organism defense from cancerous cells.

**Figure 3 antioxidants-12-01155-f003:**
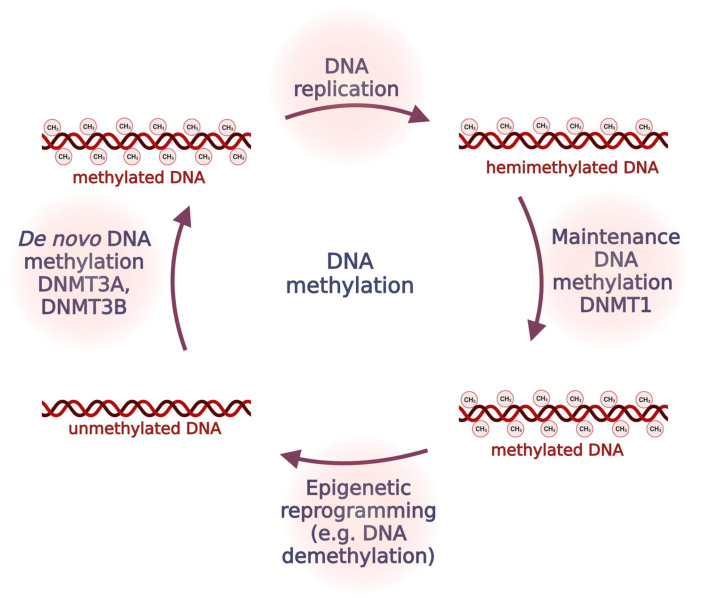
DNA methylation processes. Unmethylated DNA can undergo de novo methylation process via DNMT3A and DNMT3B. During the replication process, methylation marks are restored by DNMT1 through maintenance DNA methylation process. Abbreviations: DNMT1, DNA methyltransferase 1; DNMT3A, DNA methyltransferase 3A; DNMT3B, DNA methyltransferase 3B.

**Figure 4 antioxidants-12-01155-f004:**
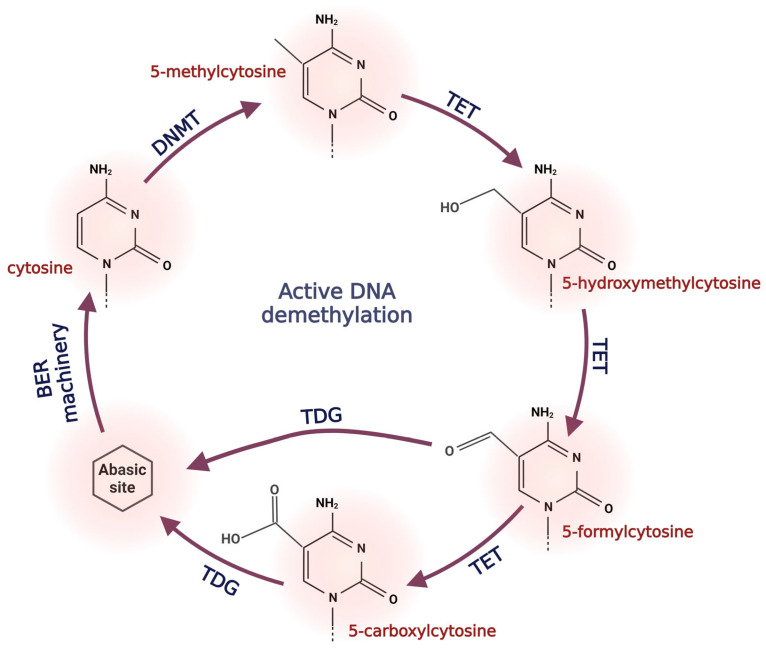
Active DNA demethylation process. 5-methylcytosine is transformed into 5-hydroxymethylcytosine, which is further hydroxylated into 5-formylcytosine and 5-carboxylcytosine. Modified cytosines (5-formylcytosine and 5-carboxylcytosine) are recognized and excised by thymine DNA glycosylase (TDG), a key enzyme belonging to base excision repair (BER) machinery. Abbreviations: BER machinery, base excision repair machinery; DNMT, DNA methyltransferase; TET, ten-eleven translocation proteins; TDG, thymine DNA glycosylase.

**Figure 5 antioxidants-12-01155-f005:**
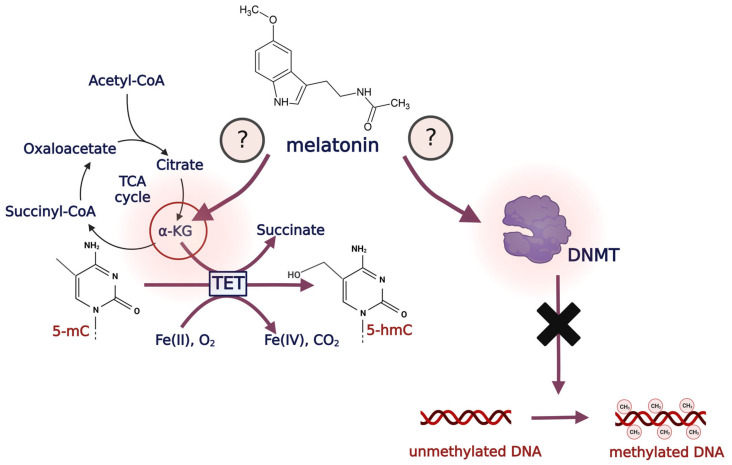
Possible impact of melatonin on DNA methylation and demethylation processes. Melatonin can impact TET proteins via the regulation of α-ketoglutarate level, thus promoting active DNA demethylation. Moreover, melatonin may also regulate DNMT, resulting in DNA methylation inhibition. Abbreviations: DNMT, DNA methyltransferase; TET, ten-eleven translocation proteins; 5-mC, 5-methylcytosine; 5-hmC, 5-hydroxymethylcytosine; α-KG, α-ketoglutarate.

## Data Availability

Not applicable.

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
