# Peer review of "Melatonin: A Potential Regulator of DNA Methylation"

_antioxidants, 2023, doi:10.3390/antiox12061155_

Round 1

Reviewer 1 Report

Melatonin in the aspect of DNA methylation seems to be a new direction of research. The work is interesting, divided into appropriate subsections. I propose to add a table ordering the type of cancer, genes related to the methyl-non-gland regulation of DNA methylation, the mechanism of melatonin formation, clinical trials, ref. The downside of the work is the duplication of information that is published in the work of Davoodvandi A et al. 2022.

Author Response

Dear Reviewer,

thank you kindly for your time and review our manuscript. Please find
our point-by-point response to the feedback we received (please see the attachement). 

Sincerely yours,

Kinga Linowiecka

Author Response file

Reviewer 2 Report

Dear authors,

I really enjoyed reading and reviewing this interesting work that describes and untangles a tremendously important theme in molecular biology, the DNA methylation process. At it has huge impact on physiological and pathological extensive domains, it has to be profoundly investigated in conjunction with all environmental factors that transform their input into epigenetic patterns further replicated and modulated into endogenous signals that abruptly change the physiologic pathways at different levels.

I consider it important to add a few personal suggestions in order to increase the scientific soundness of this review and bring it to a more complex form in order to approach a higher medical relevance for it:

-the endogenous synthesis of  melatonin should be detailed, presented in a schematic representation, with all enzymes relevant in this process

-as the journal you submitted to is dedicated to antioxidants, and melatonin, as you also briefly mentioned, is an antioxidant molecule, I advise you to detail the antioxidant mechanism of melatonin and extract valuable information that you can also link to methylation process and epigenetic modulation

-it should  also be discussed in addition to subchapters 2.1 and 2.2  about the predilect methylation site, CpG dinucleotide islands and all the details linked with the methylation of DNA in non-CpG sites that has  been found not to influence the structure or stability of chromatin, DNA and protein interactions, or gene regulation

-as aberrant methylation of DNA has been extensively reported to be involved in a wide range of cancer types, this should be carefully approached with clinical studies from literature

-I also suggest adding to 3.2 subchapter other relevant information from previous clinical studies that implied the administration of melatonin  in order to bring supplementary scientific proofs in promoting melatonin as a methylation modulator and expand this part to a more broader view. you can also present the information in a table with more details and a better exposure.

Author Response

Dear Reviewer,

thank you kindly for your time and review our manuscript. Please find
our point-by-point response to the feedback we received (please see the attachement). 

Sincerely yours,

Kinga Linowiecka

Reviewer 3 Report

In this manuscript, the author discussed the potential role of melatonin in the modulation of DNA methylation. This topic is very interesting, but the following issues need to be addressed.

1. In keeping with the theme of antioxidants, the authors should discuss whether the effect of melatonin on DNA methylation is related to its antioxidant function. Please discuss to better fit the topic.

2. Night shift work has been linked to the hypermethylation of genes involved in circadian rhythms, which has also been linked to changes in melatonin secretion. Please discuss the relationship between melatonin, circadian rhythm, and DNA methylation.

3. Is the pathway of melatonin affecting DNA methylation related to its receptor? Please discuss which receptor is primarily responsible.

4. The author discussed the effects of ALAN on melatonin and DNA methylation in 4.3. Currently, the relationship between ALAN and melatonin secretion and disease development is of great concern. Please refer to discuss: (1) Does artificial light at night increase cancer risk by affecting DNA methylation through melatonin? (2) Is exposure to ALAN relatively safe in terms of methylation or tumors in mice whose melatonin secretion is controversial, such as C57/BL6 mice, as it may not secrete melatonin. (3) The effects of ALAN on metabolic processes have also recently received much attention. Does the effect of ALAN exposure on methylation and melatonin affect metabolic processes?

5. Please add a detailed description of Figures 1, 2, and 4.

6. It is suggested to add some new references. Most of the references cited in this article are from 5 years ago.

Author Response

(The authors gave the same response as above.)

Reviewer 4 Report

This is a thorough review of the relationship between melatonin and DNA methylation. The authors then explain the role of DNA methylation in cell physiology and pathology and recommended melatonin as a potential side medication in cancer therapy. I would like to suggest some modifications to improve its quality:

 Comments:

  1. The authors present a comprehensive literature search on the details of melatonin and its role as a regulator of DNA methylation yet lack local organization of each section. Most of the sentences of the section are individual summaries of publications and lack synthesis of an interesting theme or a dominant pathway that is expected of a review paper. Authors also could add concluding or overarching themes to the end of each paragraph.
  2. The section “Melatonin and Methylation During Cell Development and Differentiation” should expand on the role of melatonin in DNA methylation regulation in other cell types as well. Currently, it is only on the oocyte and embryo.
  3. In the section "Basic Functions of Melatonin", the review should also highlight/describe the underlying mechanism of melatonin. Currently, there is a list of mechanisms in line 52 but needs details to highlight for the readers the theme or most important pathways.
  4. In the section “Melatonin and Circadian Clock Epigenetics “The definition of epigenetics should be added to line 103. Then the review should list the different types of epigenetic alterations. Next, the authors should cite articles that show a relationship between epigenetic alterations of circadian-dependent clock genes and melatonin. 
  5. As melatonin is a well-known antioxidant, the authors should expand on the antioxidant effect of melatonin and its underlying mechanisms.
  6. Based on the literature melatonin is an anti apoptotic factor. On the other hand, apoptosis plays a role in cancer, and DNA methylation also affects genes involved in apoptosis pathways. Thus, the review should have a section on the interactions between apoptosis, cancer, and melatonin.
  7. The review should have a section on the link between oxidative stress and epigenetic alterations and possible mechanisms by which melatonin regulates their relationship. 
  8.  Lines 57, 62, 93 and 129 should be cited. 
  9. The manuscript would benefit from an editing service for improving readability and grammatical errors such as run-on sentences.

Author Response

(The authors gave the same response as above.)

Round 2

Reviewer 4 Report

May 19, 2023

Comments to the authors

This manuscript has been revised according to reviewer's comments. Thanks to authors for their efforts.